# Incidence and Risk Factors of Chest Wall Metastasis at Biopsy Sites in Patients with Malignant Pleural Mesothelioma

**DOI:** 10.3390/cancers14184356

**Published:** 2022-09-07

**Authors:** Masaki Hashimoto, Michiko Yuki, Kazuhiro Kitajima, Akihiro Fukuda, Toru Nakamichi, Akifumi Nakamura, Ayumi Kuroda, Seiji Matsumoto, Nobuyuki Kondo, Ayuko Sato, Koichiro Yamakado, Tohru Tsujimura, Seiki Hasegawa

**Affiliations:** 1Department of Thoracic Surgery, Hyogo Medical University, Nishinomiya 6638501, Japan; 2Department of Molecular Pathology, Hyogo Medical University, Nishinomiya 6638501, Japan; 3Department of Radiology, Hyogo Medical University, Nishinomiya 6638501, Japan

**Keywords:** chest wall metastasis, malignant pleural mesothelioma, pleural biopsy, pleurodesis

## Abstract

**Simple Summary:**

Chest wall metastasis at biopsy site (CWM) is still an unsolved problem in malignant pleural mesothelioma (MPM) due to its aggressive invasiveness. Here we investigated the incidence and risk factor of CWM using our institutional cohort. Consecutive 262 patients who underwent curative-intent surgery following neoadjuvant chemotherapy for MPM were enrolled in this retrospective cohort study. Of 262, 237 patients were eligible. CWM was evaluated radiologically (r-CWM) and pathologically (p-CWM). Radiological examination showed r-CWM in 43 patients (18.1%), while pathological examination showed p-CWM in 135 patients (57.0%). The incidence of p-CWM was significantly higher in the patients who received pleurodesis after pleural biopsy (77.0% vs. 47.9%, <0.001). Multivariate logistic regression analysis for p-CWM revealed that pleurodesis is an independent risk factor of p-CWM.

**Abstract:**

To investigate the incidence and risk factors of chest wall metastasis (CWM) at biopsy sites in patients with malignant pleural mesothelioma (MPM). This retrospective cohort study was conducted in 262 consecutive MPM patients who underwent multimodal treatment in which including neoadjuvant chemotherapy (NAC) and curative-intent surgery, from August 2009 to March 2021. CWM was evaluated radiologically (r-CWM) and pathologically (p-CWM). We also investigated the risk factors of p-CWM and the consistency between r-CWM and p-CWM. Of 262 patients, 25 patients were excluded from analysis due to missing data or impossibility of evaluation. Of the eligible 237 patients, pleural biopsy was performed via video-assisted thoracoscopic surgery in 197 (83.1%) and medical thoracoscopy in 40 (16.9%). Pleurodesis was performed after pleural biopsy in 74 patients (31.2%). All patients received NAC followed by curative-intent surgery. Radiological examination showed r-CWM in 43 patients (18.1%), while pathological examination showed p-CWM in 135 patients (57.0%). The incidence of p-CWM was significantly higher in the patients who received pleurodesis after pleural biopsy (77.0% vs. 47.9%, <0.001). Multivariate logistic regression analysis for p-CWM revealed that pleurodesis is an independent risk factor of p-CWM (adjusted hazard ratio, 3.46; 95% confidence interval, 1.84–6.52, <0.001). CWM at the biopsy site was pathologically proven in more than half of the patients (57.0%) who received NAC followed by curative-intent surgery, which was higher than the numbers diagnosed by radiological examinations (p-CWM: 57.0% vs. r-CWM: 18.1%). Pleurodesis after pleural biopsy is an independent risk factor of p-CWM.

## 1. Introduction

Malignant pleural mesothelioma (MPM) is a highly aggressive neoplasm arising from the parietal pleura. The treatment of MPM is still challenging due to its diffuse growth with considerable invasiveness [1,2]. Ipsilateral pleural effusion is the most frequently observed type in MPM patients. As done for other malignant pleural diseases, pleural effusion cytology is performed in most cases prior to definitive diagnosis [3,4,5,6]. As the guidelines mention that definitive diagnosis should be performed through pathological examination with sufficient material, pleural biopsy is frequently performed to obtain the definitive diagnosis [6,7,8]. 

Since MPM easily infiltrates into surrounding organs due to its invasiveness, chest wall metastasis (CWM) is highly frequently observed at the biopsy site or drainage tract [9,10,11]. CWM could cause chest pain and lead to the impairment of the patient’s quality of life and performance status (PS) [12,13]. Moreover, CWM is a potentially negative predictive factor in patients who have undergone lung-sparing surgery [9,10]. However, pathological evaluation of CWM at biopsy sites has been performed in only a few reports [9,10], and there are no literature reports of the risk factors for developing CWM. Herein, we report our findings of a retrospective cohort study conducted to investigate the incidence and risk factors of CWM biopsy sites in patients who underwent curative-intent surgery for MPM.

## 2. Material and Methods

### 2.1. Ethics

The institutional review board at the Hyogo College of Medicine (number 3925) approved this study on 25 November 2021. Due to the retrospective cohort study, the requirement for written informed consent for study participation was waived by the institutional ethics committee. Instead, participants were given an opportunity to opt-out if they did not want their information to be used in this study. This study was conducted in accordance with the principles of the Declaration of Helsinki.

### 2.2. Study Design and Patient Selection

We conducted a historical cohort study to investigate the incidence and risk factors associated with CWM in MPM patients who underwent curative-intent surgery following neoadjuvant chemotherapy (NAC). Between August 2009 and March 2021, 262 consecutive patients with histologically proven MPM, who underwent multimodal treatment (MMT) at the Hyogo College of Medicine Hospital were enrolled in this study. All patients were histologically diagnosed as MPM through pleural biopsy. The pleural biopsy was basically performed via video-assisted thoracoscopic surgery (VATS) or medical thoracoscopy. Patients without pathological evaluation of the biopsy sites and those diagnosed by needle biopsy were excluded from analysis, the latter due to the impossibility of identifying the biopsy site.

### 2.3. Therapeutic Strategy

As mentioned in the therapeutic guidelines [7], curative-intent surgery should be performed as part of MMT in patients with early-stage MPM. The inclusion criteria of MMT in our institution are previously reported in some literatures [14,15,16]. The principle are as follows: (1) histologically proven MPM; (2) radiologically resectable tumor following NAC (yield clinical [yc-]stage, T0-3N0-1M0); (3) administration of MMT being feasible with a favorable performance status (PS); (4) being tolerant to curative-intent surgery; and (5) unproven N2-3.

Overall, NAC was performed in all patients who met the inclusion criteria prior to curative-intent surgery. After the completion of NAC, radiological examinations such as contrast-enhanced computer tomography (CT) and/or positron emission tomography–computer tomography (PET–CT) were performed to evaluate the response to NAC on the basis of the modified Response Evaluation Criteria in Solid Tumors (mRECIST) [17]. Curative-intent surgery, including lymph node dissection, was performed in patients who showed no apparent tumor progression after NAC. 

An en bloc resection, including the biopsy site, was routinely performed [14,15,16] when curative-intent surgery was attempted (Figure 1A–C). Extrapleural pneumonectomy (EPP) was the only surgical procedure performed before September 2012 [14,15]. After September 2012, we attempted to perform pleurectomy/decortication (P/D) in all patients, and converted to EPP was only performed if macroscopic complete resection could be achieved in EPP [14,15]. After the completion of curative-intent surgery, adjuvant radiation therapy was delivered in patient underwent EPP, and adjuvant chemotherapy was delivered in the patient underwent P/D. Prophylactic radiation therapy (RT) to the chest wall after diagnostic or therapeutic interventions was not performed routinely. 

### 2.4. Evaluation of Chest Wall Metastasis at Biopsy Site and/or Drainage Tract

We evaluated the presence of CWM through not only radiological examinations but also pathological examinations. Radiological CWM at the biopsy site (r-CWM) was radiologically evaluated prior to curative-intent surgery using contrast-enhanced CT and/or PET. We defined r-CWM as the following situations: on contrast-enhanced CT images, lesions were diagnosed when an apparent tumor nodule was detected at the biopsy site. On PET–CT images, lesions were diagnosed when abnormal focal fluorodeoxyglucose uptake relative to the uptake in comparable normal structures or surrounding tissue—observed on PET–CT images—corresponded to an abnormal mass on CT. The diagnosis of r-CWM was performed by a diagnostic radiologist (KK).

Pathological CWM at the biopsy site (p-CWM) was evaluated using hematoxylin-eosin stain and immunohistochemical stains (Figure 2A–E). We confirmed as positive for p-CWM when AE1/AE3-positive atypical cells are invading into the subcutaneous tissue at the biopsy site regardless tumor size or number of the atypical cells. The diagnosis of p-CWM was performed by pathologists (TT and MY).

### 2.5. Data Collection

Medical records of all enrolled patients, including operation notes, radiological findings, and pathological findings, were collected. Clinical and pathological stages were determined using the Eighth Edition of Tumor Node, Metastases Classification, proposed by the International Mesothelioma Interest Group and the International Association for the Study of Lung Cancer [18,19,20]. Overall survival (OS) rates were calculated from the time of the first pathological diagnosis to the date of the most recent follow-up or death.

### 2.6. Statistical Analysis

The primary endpoint of this study was to investigate the incidence of CWM at the biopsy site in MPM patients who underwent curative-intent surgery following NAC. The secondary endpoints were to investigate the risk factors of CWM and investigate the consistency between r-CWM and p-CWM.

Fisher’s exact test was employed to compare between the two groups. Logistic regression analysis was utilized to estimate the associations of the patients’ baseline characteristics with p-CWM. Survival was analyzed using the Kaplan-Meier method, and comparison of the survival rates between the 2 groups was performed using the log-rank test. All statistical analyses were performed using EZR (Saitama Medical Center, Jichi Medical University, Saitama, Japan), which is a graphical user interface for R (The R Foundation for Statistical Computing, Vienna, Austria). Two-tailed *p* values < 0.05 were considered statistically significant.

## 3. Results

### 3.1. Study Flowchart and Patient Characteristics

Two hundred and sixty-two consecutive patients were enrolled in this study. Of 262 patients, 14 patients were excluded from analysis due to the absence of pathological examination of the biopsy site, and 11 patients were excluded due to diagnosis through needle biopsy. Therefore, the analyses of this study were performed in the remaining 237 patients. Patient characteristics are shown in Table 1. The median age was 67 years, right-side involvement was observed in 136 patients (57.4%), and there were 193 males (81.4%). Definitive diagnosis of MPM was obtained through video-assisted thoracoscopic surgery (VATS) pleural biopsy in 197 (83.1%) and medical thoracoscopy in 40 (16.9%) patients. Prophylactic irradiation prior to NAC was performed in only 2 patients. Pleurodesis was performed in 74 patients (31.2%). All patients received NAC, and apparent tumor progression was not seen in radiological examination after NAC. EPP was performed in 47 (19.8%), P/D in 170 (71.7%), and exploratory thoracotomy in 20 (8.4%).

### 3.2. Evaluation of CWM and Diagnostic Accuracy of Radiological Examination for p-CWM

Radiological examination through contrast-enhanced CT and/or PET–CT showed r-CWM in 43 patients (18.1%), while pathological examination showed p-CWM in 135 patients (57.0%). The correlations between radiological findings and p-CWM are shown in Table 2. The sensitivity and specificity of radiological examination were 23.0% and 88.2%, respectively. The positive predictive value and negative predictive value were 72.1% and 46.4%, respectively.

### 3.3. Risk Factors of p-CWM

The correlations between patient characteristics and p-CWM are shown in Table 3. In brief, p-CWM was more frequently observed in males (59.6% vs. 45.5%, *p* = 0.09), on the right side (61.0% vs. 51.5%, *p* = 0.15), and was diagnosed through VATS pleural biopsy (58.9% vs. 47.5%, *p* = 0.22). Significant correlation with p-CWM was observed in patients who received pleurodesis after pleural biopsy (77.0% vs. 47.9%, *p* < 0.001). The results of multivariate logistic regression analysis for p-CWM are shown in Table 4; pleurodesis after pleural biopsy was an independent risk factor of p-CWM (adjusted hazard ratio, 3.46; 95% confidence interval, 1.84–6.52, *p* ≤ 0.001). 

### 3.4. Prognostic Impact of p-CWM

The median follow-up duration was 30.9 months among 237 patients. Median survival time in this cohort was 40.8 months. As shown in Figure 3, survival time was significantly shorter in patients with p-CWM (49.8 months vs. 37.4 months, *p* < 0.05). In multivariate analyses, p-CWM tended to be correlated with poor prognosis.

## 4. Discussion

This cohort study revealed two insights into CWM in patients who underwent curative-intent surgery following NAC. First, the incidence of p-CWM was higher than r-CWM (57.0% vs. 18.1%). Second, pleurodesis was an independent risk factor associated with CWM, a finding, to our knowledge, of the first investigation analyzing the risk factors for CWM.

The incidence of CWM in this study was higher than that shown in previous reports [11,21,22,23,24]. Agarwal and colleagues investigated the correlations between the incidence of CWM and diagnostic procedures, which revealed that 23.8% with thoracotomy, 15.7% with thoracoscopy, 9.1% with chest tube drainage, and 3.6% with thoracocentesis [25] This would be the reason why most of the literature reports on CWM describe diagnoses through radiological and/or physical findings without pathological findings [10,11,22,24]. This study also showed an extremely low sensitivity of radiological examination (23.0%). As shown in Figure 2, precise pathological examination demonstrated micrometastases, which might be impossible to detect in radiological examinations. Even though radiological examination, especially PET–CT, could help clarify the tumor status [21], this cohort study suggested that CWM may be present at the biopsy site or drainage tract regardless of the radiological findings.

Given the higher incidence of CWM found in this study, the diagnostic procedure have to be considered with its risk and benefit. Although it is preferable to obtain a sufficient amount of material, an all-layer pleural biopsy should be done only at the incision site. Several literature reports reveal that loco-logical recurrence is the most frequent recurrence pattern in MPM patients who underwent curative-intent surgery [26,27]. According to these results, because all-layer pleural biopsy could frequently lead to CWM, pleural biopsy at multiple sites might be related to postoperative loco-logical recurrence. In fact, it is difficult to perform an extended resection of biopsy sites due to anatomical restrictions, such as the mediastinal side. In contrast, the incidence of CWM had been reported to be lower in thoracocentesis or needle biopsy than in VATS pleural biopsy [25]. Recent advances in cytological examination, including the cell block procedure, could provide reliable diagnoses in selected patients [28,29,30,31,32]. Since the latest guidelines also accept definitive diagnosis using pleural effusion cytology [33], it may be better if pleural biopsy is taken in only well-considered cases.

Surprisingly, this study revealed that pleurodesis is an independent risk factor of CWM. As recommended by the guidelines, pleurodesis is commonly performed in patients with malignant pleural effusion (MPE) [13]. Pleurodesis helps in not only controlling pleural effusion-related symptoms in MPM but also improving the prognosis [34,35]. Moreover, pleurodesis has been performed in patients who are candidates for P/D because visceral pleurectomy may be easier in patients with complete pleural adhesion [16]. However, this study revealed that pleurodesis might be associated with tumor growth, which gives rise to two hypotheses. One hypothesis is the presence of chest wall adhesion, which may make it easier for the tumor to infiltrate the chest wall. A pleural biopsy can induce the pleural defect at the biopsy site, leading to the tumor infiltration into the chest wall. Given the positive correlation between the size of the biopsy site and the incidence of CWM [25], pleurodesis could promote tumor infiltration at the biopsy site. The other hypothesis is that pleurodesis promotes inflammation, which in turn promotes tumor growth. Chemical pleurodesis such as talc pleurodesis promotes inflammation in the thoracic cavity, resulting in pleural adhesion; the inflammation may promote the tumor growth. Indeed, T-stage upstaging (yc-T < p-T) was more frequent in patients who received pleurodesis (67.6% vs. 54.6%, *p* = 0.07). An analysis of only patients who received talc pleurodesis revealed that the incidence of T-stage upstaging was significantly higher in patients with talc pleurodesis (70.8% vs. 54.8%, *p* = 0.05). Several inflammatory cytokines such as interleukin 6, tumor necrosis factor-β, and vascular endothelial growth factor could induce tumor growth of mesothelioma [36,37,38]. Thus, by promoting the release of inflammatory cytokines [39], pleurodesis might lead to tumor growth.

Talc pleurodesis is the most standardized therapeutic strategy in patients with MPE [13]. However, this study suggested that pleurodesis following pleural biopsy was an independent risk factor of CWM. Pleural effusion, a well-known symptom in MPM, is often associated with tumor progression and is often reduced or disappears after systemic chemotherapy. These results suggest that the indication of pleurodesis should be considered only in case with severe respiratory symptoms.

Prophylactic radiation therapy (RT) to the chest wall after diagnostic or therapeutic interventions has been widely performed to prevent the development of CWM [12,40,41,42]. However, the efficacy of prophylactic RT remains controversial [40,42]. In this cohort, as there were only 2 patients who underwent prophylactic RT after pleural biopsy, it was difficult to evaluate the clinical impact of prophylactic RT. Our results revealed that CWM developed in 57.0% of the cases even when NAC was administered, which suggests that CWM would also develop equally or more frequently in patients with advanced or unresectable MPM. As randomized controlled studies have shown that severe adverse events related to prophylactic RT rarely occur [12,40], prophylactic RT may be considered to prevent the development of CWM in patients with unresectable or advanced MPM. 

Bölükbas and colleague showed the presence of CWM was significant poor prognostic factor in patients who underwent curative-intent surgery [9]. Richards and colleague also showed the tumor invasion at previous chest tube had a negative survival impact and proposed it as T4 status [10]. As in the previous reports [9,10], p-CWM tended to be correlated with a shorter prognosis, however not significant. In this study, p-CWM was observed in more than half of the patients in this study, which was much higher than Richards’s literature (57.0% vs. 10.0%). This would be a reason for this discrepancy between this study and previous reports on why the minimal-CWM patients were included in this study as shown in Figure 2. 

Several limitations have to be considered in this study design. First, this study was a single-institution retrospective study with selected patients who had undergone MMT. In our therapeutic strategy, curative-intent surgery was not indicated for patients with apparent tumor progression after NAC. Since MMT was not indicated for most of the MPM patients, the incidence of CWM should be investigated in histologically proven MPM patients regardless of their tumor status. Second, the procedure of pleural biopsy could not be standardized due to the retrospective nature of the study. Third, although this study revealed that pleurodesis was an independent risk factor of CWM, the indication and the agents used were also not standardized. Therefore, a prospective study must be conducted to evaluate the association between tumor progression and pleurodesis. Fourth, the grading of p-CWM according to size were not evaluated in pathological examination. It might be caused survival impact of CWM was not observed in this study.

## 5. Conclusions

We investigated the incidence and risk factors of CWM. CWM was pathologically proven in more than half of the patients, which was higher than the number diagnosed by radiological examinations (57.0% vs. 18.1%). Pleurodesis was found to be an independent risk factor of CWM after pleural biopsy. Further investigation must be conducted to clarify the correlation between pleurodesis and tumor progression.

## Figures and Tables

**Figure 1 cancers-14-04356-f001:**
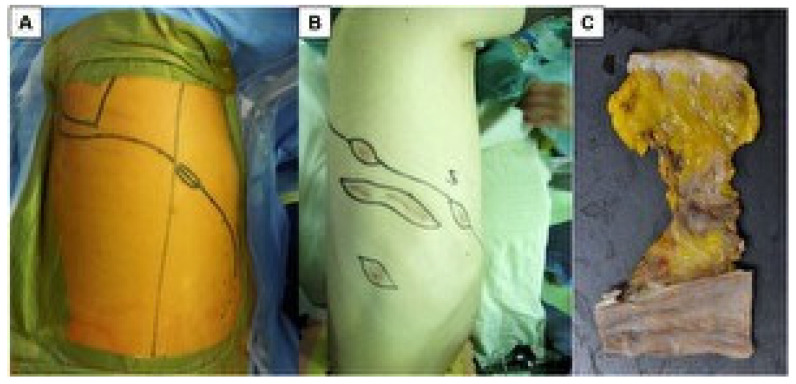
En bloc resection was attempted as curative-intent surgery. When pleural biopsy was performed via a single port, a posterolateral incision, including the biopsy site, was made (**A**). When pleural biopsy was performed via multiple ports or incisions, a posterolateral incision and an isolated incision-like island were made (**B**). Isolated removed biopsy site in which was removed including skin to parietal pleura (**C**).

**Figure 2 cancers-14-04356-f002:**
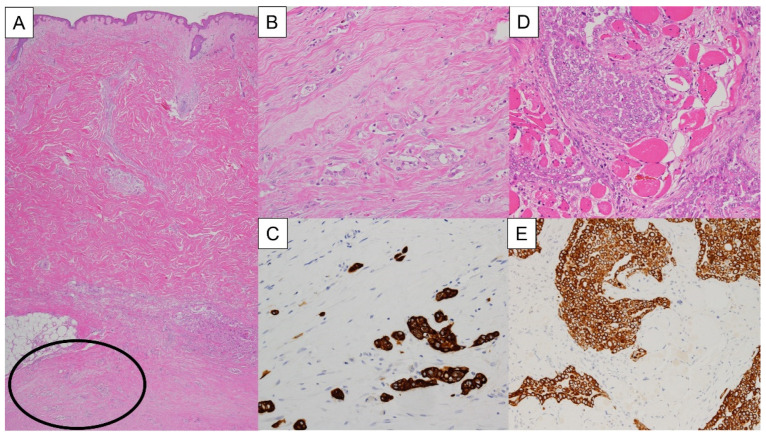
Histopathology of biopsy sites of the chest wall. Atypical cells are invading into the subcutaneous tissue at the biopsy site (circle) (**A**). An enlarged image of the circle in Figure 2A (**B**). AE1/AE3 immunostaining of a section adjacent to Figure 2B, indicating that.AE1/AE3-positive atypical cells, mesothelioma cells, are invading into the subcutaneous tissue at the biopsy site as small cellular aggregates (**C**). Atypical cells are invading into the skeletal muscle of the chest wall at the biopsy site (**D**). AE1/AE3 immunostaining of a section adjacent to Figure 2D, indicating that AE1/AE3-positive atypical cells, mesothelioma cells, are invading into the skeletal muscle of the chest wall at the biopsy site as cell clusters (**E**). (**A**,**B**,**D**) are stained with hematoxylin and eosin. (**A**–**C**) are histopathological images of one patient, and (**D**,**E**) are histopathological images of another patient.

**Figure 3 cancers-14-04356-f003:**
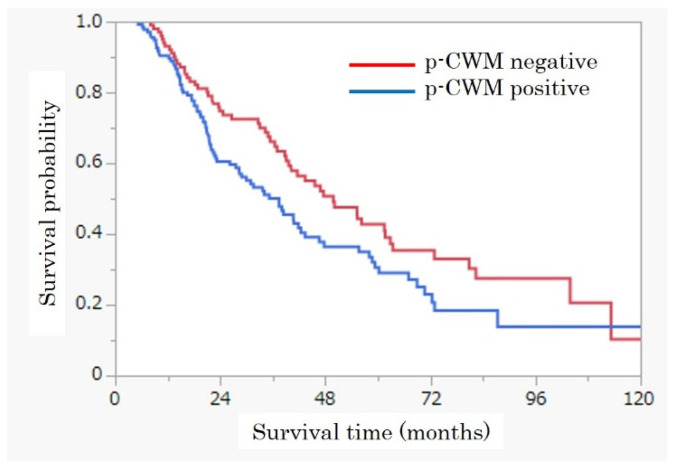
Survival curve according to p-CWM, which showed that the patients with p-CWM had a significant shorter survival time than without p-CWM.

**Table 1 cancers-14-04356-t001:** Patients’ characteristics.

Age	Median (Range)	67 (16–79)
sex	female	44
	male	193
PS	0	217
	1	20
side	left	101
	right	136
histology	epithelioid	211
	non-epithelioid	26
yc-T factor	1/2/3	145/43/49
diagnostic modality	VATS	197
	medical thoracoscopy	40
prophylactic radiation therapy	presence	2
	none	235
pleurodesis	yes	74
	no	163
r-CWM	positive	43
	negative	194
surgical procedure	EPP	47
	P/D	170
	exploratory	20
response of NAC	PR	43
	SD	194

**Table 2 cancers-14-04356-t002:** Correlations between radiological findings and p-CWM.

		p-CWM	
		Positive	Negative
r-CWM	Positive	31	12
	Negative	104	90

**Table 3 cancers-14-04356-t003:** Univariate analyses between p-CWM and clinical variables.

Variables		Positive	Negative	*p*-Value
Histological type	Epithelioid	119	92	0.68
	non-epithelioid	16	10	
Pleurodesis	present	57	17	<0.001
	none	78	85	
Prophylactic RT	present	1	1	1
	none	134	101	
Procedure of pleural biopsy	VATS	116	81	0.22
	medical thoracoscopy	19	21	
Response of NAC	PR	21	22	0.24
	SD	114	80	
yc-T	yc-T1	85	60	0.59
	yc-T2-3	50	42	
PS	0	123	94	0.82
	1	12	8	
Sex	male	115	78	0.09
	female	20	24	
Side	right	83	53	0.12
	left	52	51	

**Table 4 cancers-14-04356-t004:** Multivariate analyses between p-CWM and clinical variables.

		OR	95%CI	*p*-Value
yc-T	ycT2-3	0.99	0.70–1.41	0.97
pleurodesis	present	3.46	1.84–6.52	<0.001
NAC	SD	1.3	0.63–2.68	0.48
histology	non-epithelioid	1.33	0.55–3.22	0.53
procedure	VATS	1.28	0.63–2.61	0.37

## Data Availability

The data presented in this study are available in this article.

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
