# Peer review of "Incidence and Risk Factors of Chest Wall Metastasis at Biopsy Sites in Patients with Malignant Pleural Mesothelioma"

_cancers, 2022, doi:10.3390/cancers14184356_

Round 1
Reviewer 1 Report
This is a well-written manuscript reporting a retrospective single-center study on factors impacting on the frequency of chest wall metastasis at the biopsy site (CWM) in a quite extended cohort of MPM patients eligible for curative-intent surgery. The data from pathological evaluation suggest that by radiological evaluation is underestimating the incidence of CWM and the pleurodesis is strongly promoting CWM.
The study results are interesting and worth publishing after a few revisions. 1) Why is the impact of CWM and the other factors like pleurodesis on progression-free and overall survival of the MPM patients not analyzed? The authors suggest from literature that pleurodesis is prolonging survival. But if it supports CWM, this should be rather inverse. The survival data should be included or, if not, explained why they are not!
2) Generally, it is questionable whether these few tumor cells as shown in Figure 2 can indeed drive the disease progression. Is there any possibility to find out an impact of minimal CWM on the further course of disease in this retrospective cohort? This aspect should be addressed in the discussion section.
Author Response
Comments and Suggestions for Authors
This is a well-written manuscript reporting a retrospective single-center study on factors impacting on the frequency of chest wall metastasis at the biopsy site (CWM) in a quite extended cohort of MPM patients eligible for curative-intent surgery. The data from pathological evaluation suggest that by radiological evaluation is underestimating the incidence of CWM and the pleurodesis is strongly promoting CWM.
The study results are interesting and worth publishing after a few revisions.
1) Why is the impact of CWM and the other factors like pleurodesis on progression-free and overall survival of the MPM patients not analyzed? The authors suggest from literature that
pleurodesis is prolonging survival. But if it supports CWM, this should be rather inverse.
The survival data should be included or, if not, explained why they are not!
àThank you for your comments. Actually, we analyzed the survival impact of CWM. However, significant survival impact was only observed in univariate analysis, not in multivariate analysis. Therefore we considered that the survival impact of CWM should not be address in this manuscript due to leading misunderstanding.
Although we showed several references suggesting that pleurodesis is improving survival, these studies focused on patients without curative-intent surgery as a part of multimodal treatment. Pleurodesis would provide symptom relief for these patients, which would contribute to prolonging their survival. Based on the above reasons, the situation of this study design was different from these references.
2)Generally, it is questionable whether these few tumor cells as shown in Figure 2 can indeed drive the disease progression. Is there any possibility to find out an impact of minimal CWM on the further course of disease in this retrospective cohort? This aspect should be addressed in the discussion section.
Thank you for your comments. As mentioned above, the evaluation of the grade of CWM was not performed in this study. As you mentioned, we added to mention this issue in the discussion section
Reviewer 2 Report
"Incidence and risk factors of chest wall metastasis at biopsy sites in patients with malignant pleural mesothelioma" is an original article about a never-really proved issue: the neoplastic dissemination of mesothelioma along the biopsy tract. Overall, the study is interesting and well-designed, and the results are clearly stated. The authors found a significant number of chest wall metastases (CWM) at biopsy sites.
I have some observations:
1) The Authors should clearly state the procedure of pleural biopsy (actually reported in Table 3) in the Materials and methods section.
2) The Authors should explain what is known about the topic at present. What is the rate of chest wall metastases at biopsy sites in patients with malignant pleural mesothelioma, as reported in other studies?
3) As the number of chest wall metastases at biopsy sites in this study is so high, figures have to be clearly interpretable, demonstrating the diagnosis of metastasis. The figures are not of sufficient quality, and I suggest replacing both.
4) Authors should state more details about histological diagnosis of metastases at biopsy sites. Which immunohistochemistry was performed? Were metastases always constituted by small cellular aggregates as shown in Figure 2?
5) I suggest redesigning completely the Figure 2 (that is of poor quality too), using high power filed histological images. Why to use pan-cytokeratin? It should better to use more specific mesothelial markers, like WT1 or calretinin.
Author Response
Comments and Suggestions for Authors
"Incidence and risk factors of chest wall metastasis at biopsy sites in patients with malignant pleural mesothelioma" is an original article about a never-really proved issue: the neoplastic dissemination of mesothelioma along the biopsy tract. Overall, the study is interesting and well-designed, and the results are clearly stated. The authors found a significant number of chest wall metastases (CWM) at biopsy sites.
I have some observations:
1) The Authors should clearly state the procedure of pleural biopsy (actually reported in Table 3) in the Materials and methods section.
àThank you for your comments. As you mentioned, we added to the state of procedure of pleural biopsy in the Material and methods section.
2) The Authors should explain what is known about the topic at present. What is the rate of chest wall metastases at biopsy sites in patients with malignant pleural mesothelioma, as reported in other studies?
àThank you for your comment. We mentioned the previous reports including the actual rate of CWM in the Discussion section.
3) As the number of chest wall metastases at biopsy sites in this study is so high, figures have to be clearly interpretable, demonstrating the diagnosis of metastasis. The figures are not of sufficient quality, and I suggest replacing both.
àThank you for your comment, we replaced the figures to be clearly understood.
4) Authors should state more details about histological diagnosis of metastases at biopsy sites. Which immunohistochemistry was performed? Were metastases always constituted by small cellular aggregates as shown in Figure 2?
à The histopathology of metastases at biopsy sites was generally reflected that of the primary tumor. The demonstration of tissue invasion (eg, visceral pleural/lung, parietal pleura/chest wall, among others) is a key feature in the diagnosis of malignant mesothelioma, and invasion may be highlighted with immunostains, such as pancytokeratin or calretinin (Husain et al. Arch Pathol Lab Med. 142:89-108, 2018). On the other hand, it has been reported that the cytokeratin-positive rate is 93%, while the calretinin-positive rate is 31%, in sarcomatoid mesotheliomas (Klebe et al. Modern Pathol. 23:470-479, 2010), and that sarcomatoid mesotheliomas are generally positive with pancytokeratin antibodies including AE1/AE3, but mesothelioma markers including WT1 are relatively insensitive (WHO Classification of Tumours, 5th edition, 2021). Therefore, we focused on pancytokeratin, AE1/AE3, immunostaining to examine metastases of tumor cells at biopsy sites of the chest wall.
- As shown in redesigned Figure 2, tumors metastasized as small cellular aggregates or as cell clusters.
5) I suggest redesigning completely the Figure 2 (that is of poor quality too), using high power filed histological images. Why to use pan-cytokeratin? It should better to use more specific mesothelial markers, like WT1 or calretinin.
à Based on the Review's comment, we redesigned the Figure 2, including high power filed histological images, to clarify metastatic tumor cells at biopsy sites. As mentioned in 4), the demonstration of tissue invasion is important in the diagnosis of mesothelioma, and invasion is highlighted with pancytokeratin, AE1/AE3, immunostaining. Sarcomatoid mesotheliomas are generally positive for pancytokeratin, AE1/AE3, but mesothelioma markers including calretinin and WT1 is relatively insensitive. Therefore, we used exclusively pancytokeratin, AE1/AE3, immunostaining to examine metastasis of tumor cells at biopsy sites of the chest wall.
Round 2
Reviewer 1 Report
The revision improved the paper! However, I do not understand why the authors do not include the survival time information mentioned in their response. They show a lot of parameters in univariate analysis. And if CWM is not an independent parameter in multivariate analysis, this is even more interesting and should be discussed. There would even be the possibility to calculate an interaction term e.g. of CWM and pleurodesis. After inclusion, the paper can be accepted for publication.
Author Response
Thank you for your kind comments. As you recommended, we also mentioned the prognostic impact of p-CWM with the survival curve.
